# Experience Gained When Using the Yuneec E10T Thermal Camera in Environmental Research

Adam Młynarczyk [1,*], Sławomir Królewicz [1], Monika Konatowska [1,2] and Grzegorz Jankowiak [1]

1   Faculty of Geographical and Geological Sciences, Adam Mickiewicz University in Poznań, Bogumiła Krygowskiego 10, 61-680 Poznań, Poland; slawomir.krolewicz@amu.edu.pl (S.K.); monika.konatowska@up.poznan.pl (M.K.); grzegorz.jankowiak@amu.edu.pl (G.J.)
2   Department of Forest Sites and Ecology, Faculty of Forestry and Wood Technology, Poznań University of Life Sciences, Wojska Polskiego 71D, 60-625 Poznań, Poland
*   Correspondence: adam.mlynarczyk@amu.edu.pl

**Abstract:** Thermal imaging is an important source of information for geographic information systems (GIS) in various aspects of environmental research. This work contains a variety of experiences related to the use of the Yuneec E10T thermal imaging camera with a $320 \times 240$ pixel matrix and 4.3 mm focal length dedicated to working with the Yuneec H520 UAV in obtaining data on the natural environment. Unfortunately, as a commercial product, the camera is available without radiometric characteristics. Using the heated bed of the Omni3d Factory 1.0 printer, radiometric calibration was performed in the range of 18–100 °C (high sensitivity range–high gain settings of the camera). The stability of the thermal camera operation was assessed using several sets of a large number of photos, acquired over three areas in the form of aerial blocks composed of parallel rows with a specific sidelap and longitudinal coverage. For these image sets, statistical parameters of thermal images such as the mean, minimum and maximum were calculated and then analyzed according to the order of registration. Analysis of photos taken every 10 m in vertical profiles up to 120 m above ground level (AGL) were also performed to show the changes in image temperature established within the reference surface. Using the established radiometric calibration, it was found that the camera maintains linearity between the observed temperature and the measured brightness temperature in the form of a digital number (DN). It was also found that the camera is sometimes unstable after being turned on, which indicates the necessity of adjusting the device's operating conditions to external conditions for several minutes or taking photos over an area larger than the region of interest.

**Keywords:** low-cost thermal camera; thermal imaging; remote sensing; UAV

## 1. Introduction

Remote sensing in the thermal range is based on the measurement of electromagnetic radiation emitted from the surface of an object. All objects with a temperature above absolute zero emit radiation, and the amount of energy returned is a function of the emissivity and temperature of the object's surface [1]. Thermal imaging with a digital camera is a complicated job and should be performed by a well-trained operator [2]. The quantitative interpretation of the results, including the estimation of the surface temperature of the object, may be difficult due to many factors influencing the registration, such as air temperature and humidity, wind speed, distance from objects, recording time and sensor characteristics [3–5]. In practice, distance as an image recording parameter varies from a few millimeters or centimeters to hundreds of kilometers (thermal sensors on satellite platforms, such as Sentinel-3, Landsat series or Terra/Aster).

The emission of electromagnetic radiation is related to a wide range of wavelengths of 3–35 μm. In practice, the range of 8–14 μm is most often used in sensors mounted on UAVs or in ground measurements [6]. There are also sensors recording radiation in the shorter range of 3–5 μm. Due to the absorption of radiation by ozone in the stratosphere between 9 and 10 μm, satellite sensors operate in narrower ranges (e.g., Landsat-8 channel-10: 10.6–11.19 and channel 11: 11.5–12.51 μm). Thermal sensors are characterized by a much lower resolution of the image matrices used in comparison to cameras operating in the optical range. Typical high-end thermal imaging cameras produce images with a resolution of 640 × 512 and a refresh rate of 9–15 Hz [7,8]. A single model of a thermal sensor can be adapted to different focal lengths and sold as a separate camera model. The resolution of the thermal sensor is strongly related to its price—the larger the size of the matrix, the higher the price of the device.

An important element in expanding the application of thermal sensors is the possibility of suspending them under small UAVs and recording images from a low height above the ground, which complements other possibilities of obtaining data through contact or close distance measurement, registration from the aerial and satellite level [8–11]. Due to the combination of UAV with thermal sensors, the decision to take pictures is made in a short time between the need and the moment of taking pictures. A certain barrier to the common use of thermal sensors is their high price, which may even be several times higher than the cost of the flying platform itself when using an UAV device. Therefore, some studies are working on applying low-cost solutions [10,12–14].

Thermal cameras, which for a long time were used almost only in military applications, have begun to be widely used in many industrial fields [6]. One of the most common applications in the COVID-19 epidemic today is the use of thermal cameras to assess the temperature of people entering public facilities [15]. In archaeology, thermography can be used to identify elements hidden below the ground level of old buildings [16], to identify cropmarks [17,18], to visualize the course of old roads [19] or to analyze the state of archaeological structures under the influence of meteorological conditions [20]. One of the more serious applications of thermal imaging is by emergency services to search for missing persons in difficult-to-reach areas [21,22]. For the protection of strategically important industrial and military facilities, thermal cameras are used to detect UAVs [23]. Thermal imaging is also used in the protection of and research into the natural environment. These issues include monitoring of spontaneous fires in dry areas [24–26], pointing to places where warmer waters, including sewage, reach rivers and lakes [27], monitoring rainwater runoff in the city [28], estimating heat emissions in the area of geothermal lakes [11], imaging thermal variability of the lake surface [29,30], determining the thermal diversity of the upper forest area [31] or monitoring the life of wild animals [32,33]. Thermal imaging is widely used in construction to study the thermal integrity of buildings, which is important from the point of view of climate neutrality [13,34–36]. In urban areas, thermal imaging is used to detect and track pedestrians in terms of their behavior [37].

The discussion of numerous studies conducted with the use of thermal sensors to support sustainable agricultural production or precision farming methods is particularly noteworthy [9,38,39]. Thermal imaging in agriculture is very often associated with the use of UAVs [40]. One of the most important applications is the mapping and assessment of water resources available to plants [3], which is particularly related to the detection of water stress in plants [41–44] and the monitoring and optimization of irrigation systems [45–47]. An important branch of agriculture in which thermal imaging is used is support for viticulture [48–50]. Thermal imaging is used to address many issues, including assessing damage to cereal crops [51] and soil salinity in terms of its impact on crop growth [52]. In the processing of agricultural products, thermography is often used to control the quality of food products [53].

For various reasons, thermal images are often taken at night or in the morning, in the absence of sunlight, which makes RGB photos useless. In order to process thermal images photogrammetrically, they must have enough detail necessary to connect the images with each other. With a low resolution of the thermal sensor matrix, capturing images with an appropriate level of detail is possible at suitable flight altitudes (appropriate distance from the object) when using a large field of view. The authors' experiences so far with taking pictures with the E10T camera (matrix 320 × 256, focal length f = 4.3 mm) over various natural areas indicate that the appropriate flight altitude is approximately 100 m above ground level. The sharpness of the thermal image is also influenced by the current thermal contrast of the objects. Too low a contrast makes it impossible to identify details in the photos, which is an obstacle when assembling a uniform orthophotomap. In order to reduce the impact of wind, which lowers the target temperature and humid air, acting as a shield against infrared radiation, photos should be taken in a cloudless sky and in less windy weather conditions [44]. Obtaining good-quality thermal images is also difficult in the presence of fog or atmospheric sediments on the objects (frost and rime). The legibility of thermal images is also influenced by the speed of the aircraft in relation to the image speed. The lower the speed, the better, but the selection of the flight speed is also dependent on other factors, such as the size of the area and the number of batteries (total flight time that can be achieved with their help). In the air survey above the ground, due to the different emissivity of the photographed objects or the variability of wind conditions during shooting, internal correction of the thermal sensor is impossible to use. Figure 1 presents examples of thermal images and the corresponding photos in the visible range (RGB) of natural objects, taken with the E10T camera, vertically and horizontally. The photos show fields, meadows, river backwaters, and forests at night and during the day. The most interesting photo is the river bank, where the thermal image shows the reflection of the shore on the water surface in a way very similar to images in the visible spectrum.

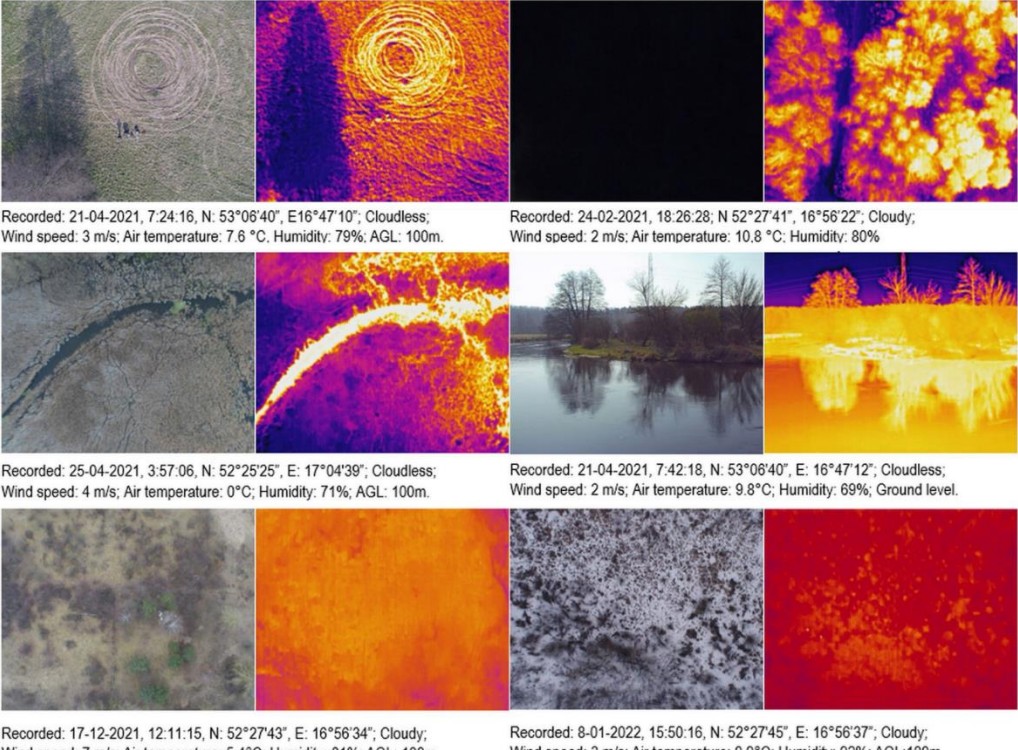

Recorded: 21-04-2021, 7:24:16, N: 53°06'40", E16°47'10", Cloudless; Wind speed: 3 m/s; Air temperature: 7.6 °C. Humidity: 79%; AGL: 100m.

Recorded: 24-02-2021, 18:26:28; N 52°27'41", 16°56'22"; Cloudy; Wind speed: 2 m/s; Air temperature: 10.8 °C; Humidity: 80%

Recorded: 25-04-2021, 3:57:06, N: 52°25'25", E: 17°04'39"; Cloudless; Wind speed: 4 m/s; Air temperature: 0°C; Humidity: 71%; AGL: 100m.

Recorded: 21-04-2021, 7:42:18, N: 53°06'40", E: 16°47'12"; Cloudless; Wind speed: 2 m/s; Air temperature: 9.8°C; Humidity: 69%; Ground level.

Recorded: 17-12-2021, 12:11:15, N: 52°27'43", E: 16°56'34"; Cloudy; Wind speed: 7 m/s; Air temperature: 5.4°C; Humidity: 81%; AGL: 100m.

Recorded: 8-01-2022, 15:50:16, N: 52°27'45", E: 16°56'37"; Cloudy; Wind speed: 3 m/s; Air temperature: 0.9°C; Humidity: 92%; AGL:100m.

**Figure 1.** Examples of various pairs of RGB and thermal images taken with the E10T camera. Below the images some information is given about the geographic and meteorological parameters (obtained from the closest station).

For a general review of thermal sensors adapted for UAV transfer, see [49]. Another review of modern thermal sensor technology in terms of autonomous aerial navigation is presented by Nguyen [54]. Thermal sensors are constructed as standalone cameras or in combination with sensors operating at other spectral ranges. Examples of such devices are the Multispectral Altum camera by Micasense and the E10 and E20 series cameras dedicated to Yuneec H520 (H520E) drones. Using a combination of different image sensors is beneficial because this provides simultaneous data registration at different ranges of electromagnetic radiation, better integration of various data, and extends the possibilities of object/surface classification, determining the state in which they are [17,18,41,48,55]. Thermal data can complement multispectral information sets, which are classified using various methods [42,46], including object classification methods [47] or methods based on machine learning algorithms and artificial intelligence [34].

One of the few thermal sensors available for the professional Yuneec H520 UAV is the E10T camera. The E10T camera was developed in cooperation with FLIR, one of the most recognizable brands on the thermal sensor market. The camera consists of a thermal sensor and an RGB sensor. The camera is available in various versions, differing in the field of view of the thermal sensor and the resolution of the thermal matrix ($320 \times 256$ and $640 \times 512$). In the cheapest version, the camera is slightly more than twice as expensive as the UAV H520 in the basic version. The thermal camera allows you to record both static images and video. It was created for various purposes, such as thermal inspection of buildings and searching for people. The Yuneec company introduced new thermal sensors for both the UAV H520 and its modified version, H520E, i.e., the E10Tv and E20Tv cameras, in 2021. The E10T camera is still available for sale, and its advantage is a very attractive price in relation to new and competitive solutions. The low-cost argument strongly justifies undertaking and continuing research on the use of this product. The camera is not officially a radiometrically calibrated device [49], i.e., it lacks the ability to perform quantitative temperature characteristic studies using the recorded images. The official user manual also does not provide much information about how the camera works. When taking pictures, it is possible to view the temperature for the center of the image and the average value for the frame, and this information is saved in the metadata in the latest version of the camera software (in previous versions it was not possible). Camera software enables a certain correction of the influence of weather conditions on the temperature indicated by the sensor. The recording of a single exposure includes an image from an RGB camera in the JPG format, a raw thermal image in the TIFF format and a thermal image in a selected contrasting color palette (in the JPG format) with increased resolution twice. However, the image in the TIFF format is a 16-bit encoded image, which would indicate that it is an image related to the actual temperature values (encoded relatively in a specific range of image numerical values, so-called DN-digital numbers). Through contact with the technical support of Yuneec (for Europe), it was found that the file plays an indirect role in creating files in the JPG format. However, the practical use of the E10T camera and the study of radiometric characteristics [56] convinced the authors of the possibilities of using this camera in quantitative environmental studies. However, such use requires full knowledge about the capabilities and features of the camera (i.e., how external conditions affect its operation).

A frequent practice in the study of the natural environment is to record thermal images in an orderly manner, i.e., photographing in the form of aerial blocks composed of parallel lines with a specific sidelap and longitudinal coverage, completed with the creation of a thermal orthophotomap, similarly to photos in the visible range. Thermal orthophotos are used as additional sources of information in various spatial information systems, for example for city management or irrigation systems. However, the photogrammetric processing of thermal images is difficult due to the lower resolution and legibility of thermal images (for the E10T sensor, the RGB camera matrix is $1920 \times 1080$ pixels, and the thermal sensor matrix is $320 \times 256$). Photogrammetric processing of thermal images may be performed simultaneously in connection with RGB images [57].

The aim of this work was to perform the radiometric calibration of the E10T camera and to analyze the statistics of thermal image sets in order to assess the stability of the E10T radiometric camera operation. A 3D printer as a heated bed to a specific temperature was used for radiometric calibration. The analysis of statistical parameters of thermal images included sets recorded over various types of land cover, taken under various weather conditions. Furthermore, thermal images were taken in vertical profiles, analyzing the variability of the reference surface image on thermal images from different heights.

## 2. Materials and Methods

### 2.1. Drone and Thermal Sensor

The research used the E10T thermal camera dedicated to the professional UAV Yuneec H520. Detailed technical data on the E10T thermal camera with an increased-sensitivity RGB camera used in the tests are presented in Table 1. During the thermal imaging of natural, agricultural and urban areas, a high setting of high contrast (high gain) was used, due to the real variability. Temperatures fall within the sensitivity range of this mode, i.e., −25 to 100 °C. The Beurer FT 65 thermometer was used to measure the temperature of the reference surfaces, for which the range of measured temperatures varies from 0 to 100 degrees Celsius [58]. The user manual of thermal camera [59] recommends not to use the camera outside the specified range of air temperatures, not to expose the sensor to strong light sources and not to take pictures in rain and high humidity (without specifying the amount of air humidity, you can guess that it is very foggy and suspended water droplets in the air).

**Table 1.** Parameters of the E10T thermal camera (Yuneec H520) [59].

| Parameter\Sensor | RGB | Thermal |
|---|---|---|
| Resolution (pixels) | 1920 × 1080 | 320 × 256 |
| Field of View (FOV) | 89.6° | 34° |
| Focal length (mm) | 3.5 | 4.3 |
| The physical dimension of a pixel (mm) | 2.3 | 12 (6 enhanced JPG) |
| Wavelength | 0.45–0.77 µm | 8–14 µm |
| Sensitivity | ISO range: 100–3200 Shutter speed: 1/30–1/8000 s | <50 mK, @f/1.0 |
| Sensor type | CMOS 1/2,8″ | Uncooled Vox microbolometer (FLIR) |
| Scene temperature range | | High gain −25 to 100 °C Low gain −40 to 550 °C |
| Calibration options | *n/a* | Atmospheric parameters: - Scene emissivity - Conversion coefficient - Atmospheric temperature |
| Color space and recording data format | RGB 24 bit, JPG | TIFF 16-bit (not radiometric), Pallete color JPEG (enhanced resolution 640 × 512) |
| Operating temperature range | −10 to 40 °C | −10 to 40 °C |

### 2.2. Radiometric Calibration

Radiometric calibration of the E10T thermal camera was performed using the heated bed of the Omni3d Factory 1.0 printer (3D printer). The software controlling the printer made it possible to manually set the temperature to which the heated bed was to be heated. The outside of the heated bed, observed and recorded by a thermal camera, is made of frosted glass. Before calibration, the 3D printer was leveled with a standard spirit level. In practice, heating the print base is necessary for stability and precise reproduction of the 3D printed object. The UAV H520 was attached to an aluminum frame placed above the printing platform as shown in Figure 2. The powered E10T camera was placed above the printing base at a height of 20 cm. In this case, it was assumed that such a distance has a negligible effect on the temperature measurement by the E10T thermal sensor. The camera was turned on for approximately 0.5 h before heating started. The verticality of the UAV H520 camera axis was set and was automatically maintained by the camera gimbal. The temperature of the printer base was varied from 18 to approximately 100 °C by heating increments of approximately 2 °C. Due to the fact that the heating process coexists with the process of external cooling of the print platform, the surface temperature was measured with a thermometer placed approximately 1 cm above the heated surface immediately before taking the thermal image, in a place corresponding to the center of the thermal image, approximately 2 min after the setting of a given temperature. Pictures with the thermal camera were taken manually using the H520 ST16s drone controller. Calibration only concerned the setting of the high gain–high sensitivity mode of the thermal sensor. The experiment was carried out in a large room with a constant temperature of around 18 °C. The calibrations were repeated three times using only the process of heating the base of the printer. In each series, the heating rate was performed according to the schedule established in the first series. After each series of measurements, the heated bed was cooled down to room temperature. On each thermal image associated with the measurement of the surface temperature of the printer base, in its center with a radius of 15 pixels, i.e., the area corresponding to the temperature field measured by the Beurer FT 65 thermometer above the print base, the DN values were averaged. The averaging was performed using the vector data model to define the range of the averaging field (circular polygon). Averaged DN values were calculated in the TNTmips version 2022 software from Landscan (US, San Luis Obispo, CA, local license for Adam Mickiewicz University). Then, the values of the temperature measured with a non-contact thermometer and the average DN values in the center of the thermal image were compared to establish the relationship and determine the functional character of this dependence. The determination of dependencies was made using the Excel software (Microsoft, US, Redmond, Office 365 license for Adam Mickiewicz University).

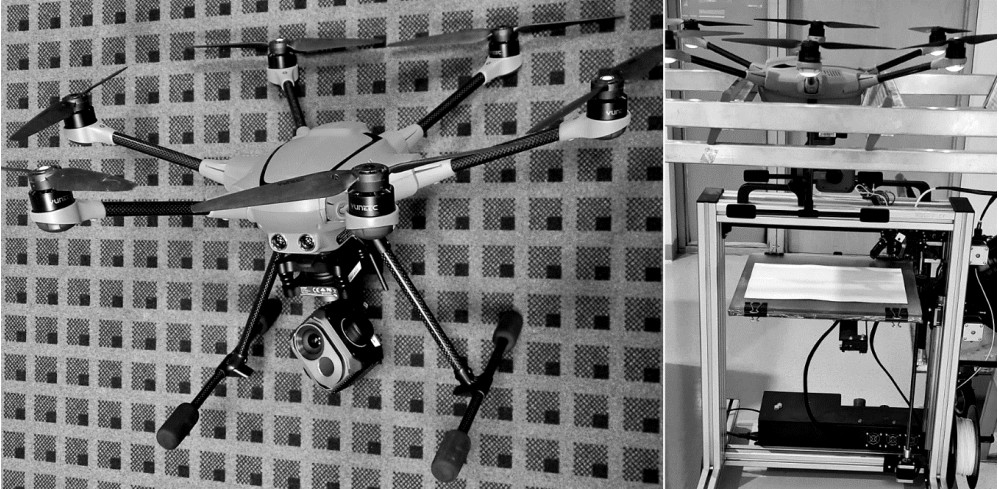

**Figure 2.** H520 drone with the E10T camera. The drone is attached to the radiometric calibration over the heated bed of the Omni3d Factory 1.0 printer.

### 2.3. Analysis of Statistical Parameters of Thermal Image Sets

For blocks of thermal images, statistical parameters were calculated on DN values such as average, minimum and maximum. Blocks of photos taken over different areas at different times of the day and seasons, using one or more batteries to complete the entire survey, were selected for analysis. These areas include (1) the area of a small stream valley, covered with scattered vegetation, marked as location-L1, (2) the separate-family housing area, marked as location L2, and (3) the area of a field covered with rape cultivation, marked as location, L3. The photos were taken according to two scenarios: (1) flight along the designated route for high lateral and longitudinal coverage, usually amounting to 80 and 85%, respectively, and the automatic image recording resulting from these settings; (2) flight along the designated route for lateral and lateral coverage with manual shooting thermal (assuming a very high longitudinal coverage and checking if it is possible to take pictures). The photogrammetric processing of thermal images in the TIFF format to the form of orthophotomaps was carried out in Agisoft's software—Metashape Professional, version 1.7.6 (Russia, St. Petersburg, local license for Adam Mickiewicz University). Standard processing steps include aligning photos (aerotriangulation, sensor autocalibration), generating a dense point cloud, calculating the surface model and orthorectifying photos, and assembling them into a continuous orthophotomap [60]. No tonal equalization methods were used when editing the photos and no vignetting effects were removed. For the visualization of the obtained orthophotos, the contrast stretching method was applied using the curve shape normalization method and the global linear stretching, based on the DN temperature range for all orthophotos.

### 2.4. Taking Photos in Vertical Profiles

The radiometric stability of the camera operation was analyzed by flying the drone over the same reference surface up and down, recording thermal images every ten meters up to a height of 120 m, while hovering (which in practice, taking into account the positioning accuracy of the UAV H520 given in the technical documentation, means absolute accuracy measuring the horizontal position up to approximately 2 m; the height above the surface was determined using data from the IMU). The overall flight time up and down twice was approximately 10 min. A fragment of the car parking with a uniform paving stone cover was selected as the reference area. On each thermal image, the extents of the reference surface was manually determined. The extent of this surface was saved as a polygon in the vector format. Then, for the range of the reference surface in all photos, the average value of the temperature in the form of DN was determined. The calculations were performed in the TNTmips software. The average temperatures were then compared with the height of the image registration in graphs to show the relationship between altitude and temperature using Microsoft's Excel software.

## 3. Results and Discussion

During the radiometric calibration of the E10T camera with the use of the heated bed of the 3D printer, three series of measurements were carried out. Figure 3, part A, shows 45 thermal images from the first series, taken while heating the heated bed from 18 to 100 °C. For the entire set of photos, a uniform contrast was adopted using the linear method, in the full range of DN variation of the entire set of photos, in order to show temperature changes throughout the heating cycle. In the central part of each photo, the area of the actual temperature measurement with a thermometer and the calculation area of the average DN value is marked with a black circle. The thermal images in part A of Figure 3 show how the base of the print is heated, generally oval in nature, with a temperature drop towards the edges of the base of the print. To show the temperature variation in the full range (18–100 °C), a uniform contrast was set for the whole set of photos. In Figure 3, part B, the same set of photos is shown, changing the way of contrast to a method that normalizes the shape of the histogram (Gauss curve), doing this separately for each photo, which allows you to see the temperature distribution of the base of the

print within the photo. These photos also show the variability of the heating of the print base as a function of the distance from the heater. The temperature measurement in the center of the image for calibration was correct because in the averaging area (black circle in the illustration), the temperature was quite stable, not affected by heating. Figure 3, part C, shows the statistics of consecutive thermal images, the mean brightness (mean) and standard deviation (SD) of all measurement series, calculated using histograms. By repeating the pictures according to the same heating scheme, it was possible to maintain a relatively linear, uniform temperature increase in the heating process.

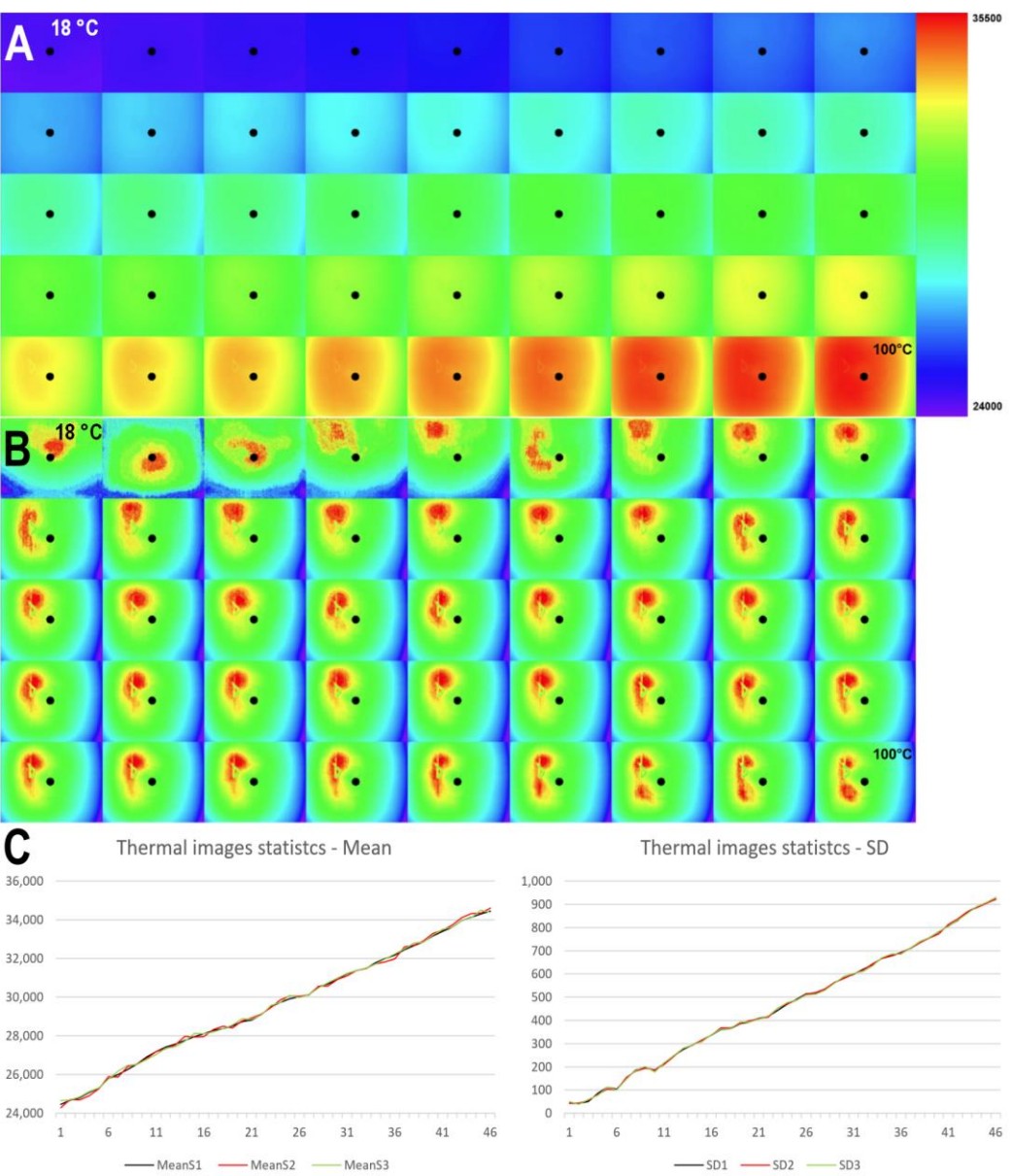

**Figure 3.** Radiometric calibration with the use of a 3D printer base: (**A**) thermal images recorded during the first calibration series, while heating the base of the 3D printer printout from 18 to 100 °C, arranging the images along with the temperature increase from the upper left corner to the lower right corner; in the central part, the area for which mean temperature values were calculated is marked with a black dot. DN values for all TIFF images recorded in 16-bit coding varied from 24,000 to 34,500; in this range, linear contrast stretching was used and visualization was with the same color palette. (**B**) The same thermal images with the changed method of contrast stretching to the method normalizing the shape of the histogram, separately for each image. (**C**) Graphs of image statistics, mean thermal brightness and standard deviation and their variability over time for three calibration series.

Figure 4 shows the results of the radiometric calibration of the E10T camera in the form of the relationship between the temperature measured over the heated bed (thermometer) of the 3D printer and the DN values recorded on the images from the thermal camera (in the TIFF format). The presented results confirm the earlier simplified results of radiometric calibration [57] that between the temperature and DN values of the thermal image it is linear and is also similar to the results presented by Kelly et al. [61]. By adopting the formula of a simple linear equation, adjusting it to the analyzed variables, namely MT = a × DN + b, the functional forms were determined using the Excel software trend line tool. For all series of measurements, the obtained slope value of the linear equation was the same, a = 0.0078, and slightly different values of the intersection with the ordinate axis (coefficients b in the linear equation equation). The differences between the measured and predicted values using the obtained relationships, presented in the graphs in the lower part of Figure 4, do not exceed two degrees. According to the authors, such differences in relation to thermal images of the natural environment are acceptable.

The radiometric calibration presented in this paper is more accurate than the previous one made by the authors [56], but is also simple and made with the use of equipment available to the authors, and not as accurate as that presented in the work of Leblanc et al. [62]. The differences in the aspect of the calibration measurements concern the distance of the thermal camera from the calibration surface and the use of two thermal points for calibration. The different distance of the thermal sensor from the calibration surface causes a difference in the linear equation coefficient describing the relationship between the temperature of the calibration surface and the DN value in thermal images, 0.0078 in the current calibration and 0.0099 in the previous calibration [56]. In the previous calibration, the distance of the sensor from the reference surface was 2 m; in the current one, it was 20 cm—the conclusion may be trivial, but the 2 m thickness of the air layer in a closed room also affects the values recorded in the images.

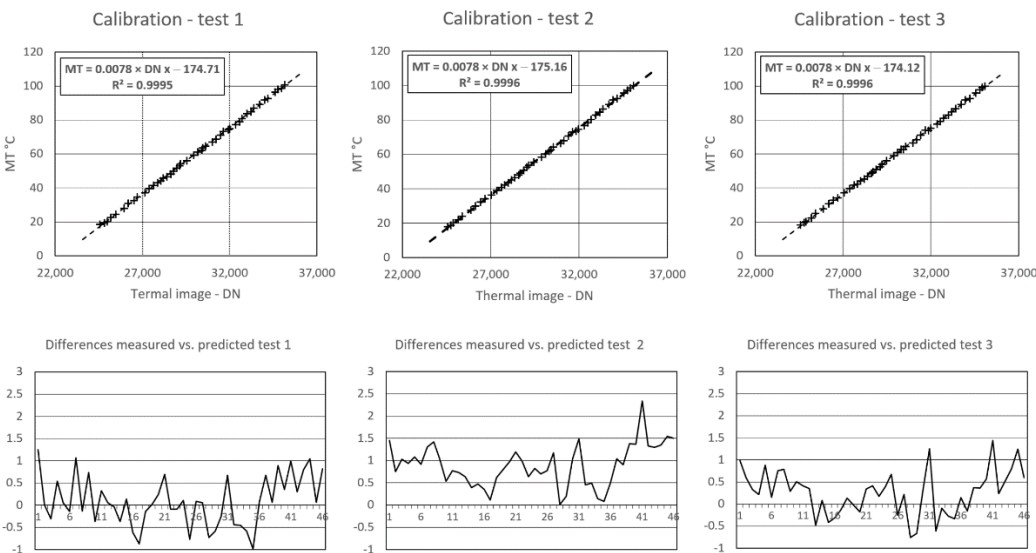

**Figure 4.** Radiometric calibration of the E10T camera obtained using the heated base of the Omni3d Factory 1.0 printer. The upper part of the illustration shows the relationship between the temperature measured on the surface of the heated base and the DN values read on the thermal images. In the lower part of the figure, the difference between the measured temperature and the one estimated using the established dependence is presented.

Figure 5 presents six thermal orthophotomaps created from images taken with the E10T camera for two locations (L1 and L2), recorded in the autumn–winter period. The photos were taken at different times of the day, visually resulting in very different readability of the orthophotos. The orthophotomap presented in part A was prepared using photos taken in the morning over a built-up area (L1) with a cloudless sky. Its legibility can be described as high and related to the differences in temperature between buildings, namely the warmest and brightest are local heat sources characteristic of single-family housing. The orthophotomap in part B was prepared using photos taken after sunset, in the period when the objects begin to reflect the heat absorbed during the day, and its legibility, the contrast between the objects, is also related to the differences in the temperature of the objects. Parts C and F present an orthophotomap made of photos recorded in the middle of the day under full cloud cover. In the C orthophotomap, the cold watercourse stands out the most (the defect formed during the installation of thermal images is also visible). In the orthophotomap F, the stream is thermally similar to the trees. Orthophotomaps D and E were made using photos recorded under direct sunlight, and therefore the orthophotomaps show thermal differences between the surfaces illuminated by the sun's rays and those located in the shade. The readability of the orthophotomap E is very similar to the image of orthophotomaps of photos taken in the visible range. Orthophotomap D, composed of photos taken closer to noon (11:55 AM), more clearly shows the differences in heating of various surfaces in relation to the topography (southern slopes are warmer, brighter). In addition, trees appear in it as bright spots.

Some of the orthophotomaps presented in Figure 5 show more general tonal differences, not related to the variation in the coverage of the photographed area. Namely, on the C, D, E, and F orthophotos, there are clear tonal differences between large fragments of the orthophotos. In the C, D, and E orthophotos, the northern part is clearly brighter (warmer) than the southern part. In the orthophotomap in part F, the northern part of the orthophotomap is dark. Orthophotomap B is tonally uniform. In orthophotos B and E, there are also visible defects related to the correct assembly of individual photos (this applies to the southern part of these orthophotos). After analyzing the order of the images, it turned out that the dark fragments of the orthophotomaps were created using photos recorded in the initial phase of the raids. This may indicate the radiometric instability of camera operation.

This is more precisely visible in Figure 6, which presents thermal images in the order of registration and with a uniform contrast setting for the entire set of photos. The illustration shows two seedlings performed directly in succession for the L3 site, which includes a rapeseed field with other crops. The photos are presented in this form because, despite many attempts, it was not possible to align the photos (aerotriangulation) and, consequently, the orthophotomaps were not created. The reasons for the impossibility of assembly are too great similarity of the content of thermal images, too low coverage between them and too low height of the flight above the ground—approximately 70 m.

To analyze more precisely and show the identified problem of radiometric instability of thermal camera operation, statistical parameters of thermal images were calculated (mean maximum and minimum). Figure 7 presents graphs of statistical parameters of photos according to the order of registration of the sets of photos presented in Figure 5 (charts A, B, C, D, E, and F) and Figure 6 (charts G and H). The instability of the thermal camera's operation concerns the initial stage of the camera's operation and is best visible on the graphs of the average brightness of subsequent photos, when the average is constantly increasing and it is not related to the temperature of the objects (it is best visible in the graphs D, E, F). In graphs G and H, graphs are presented for sets of photos taken for location L3 directly one after another—in graph H, when the device was after a longer period of operation and a short shutdown to change the battery, the initial instability is much less noticeable. A similar instability of the camera's operation is reported by Olsson et al. [63] for the multispectral Sequoia camera and initial warming is suggested.

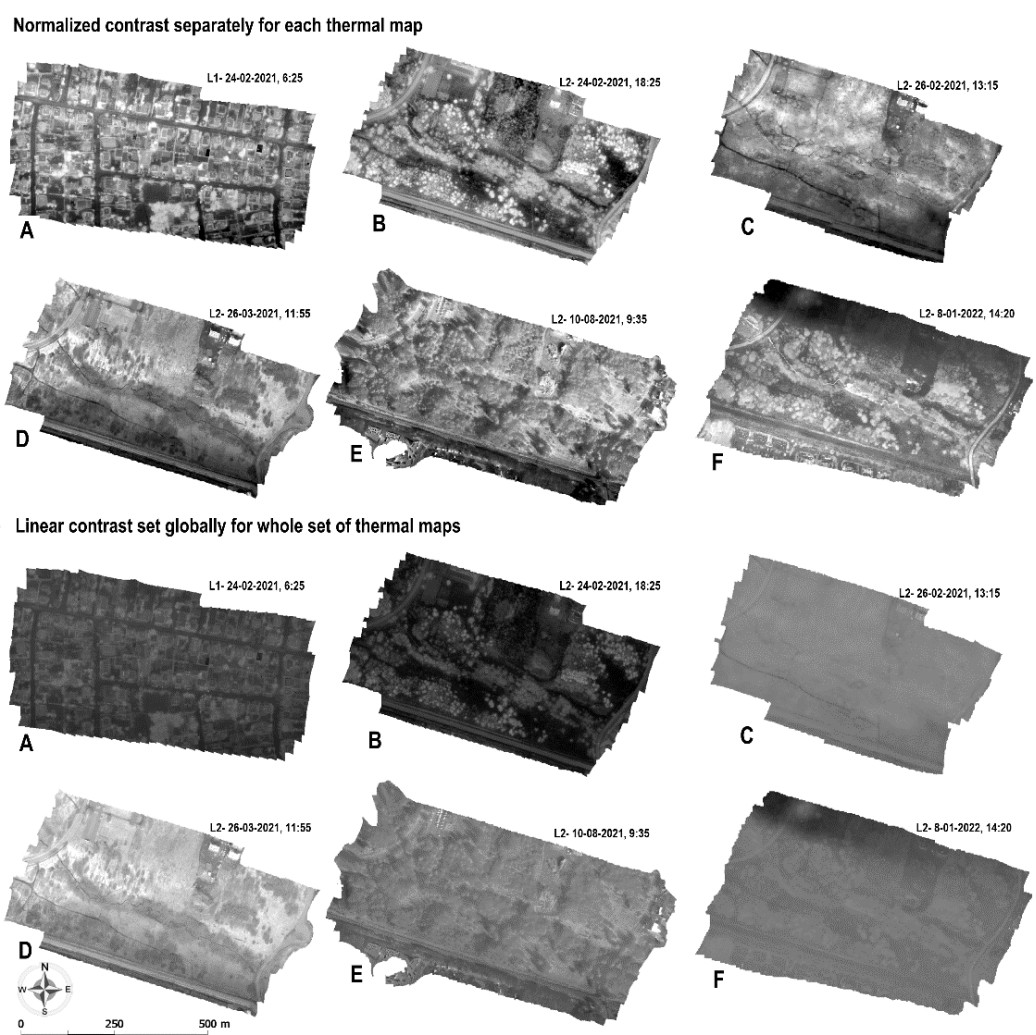

**Figure 5.** Thermal orthophotomaps computed for the sets of images taken with the E10T camera using Agisoft Metashape software for the location L1 (1 term, subfigure (**A**)) and locations L2 (five terms, subfigures (**B**–**F**)). Orthophotomaps are presented in two ways: using the contrast normalized for each orthophotomap (upper part of the illustration) and linearly defined contrast for full variation DN values of the orthophotomap set (lower part of the illustration). The pictures were taken from the height set to 100 m AGL. Orthophotos are oriented in real geographic directions.

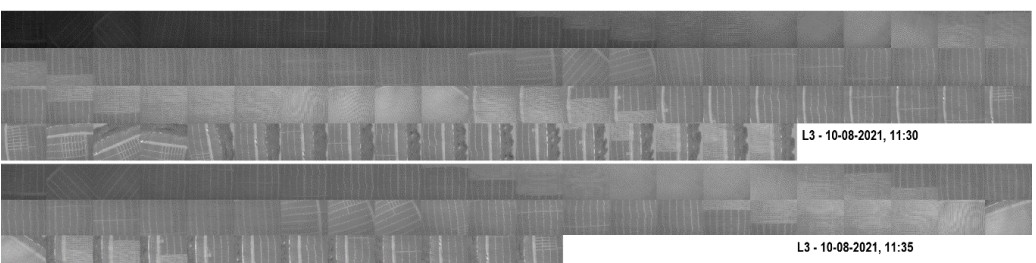

**Figure 6.** Two sets of thermal images arranged according to the order of registration for location L3 (rape field). The graphs of the statistical parameters marked with the letters G and H correspond to them in Figure 7. The same contrast table was set up for both sets using the linear method, using the minimum and maximum DN values of the entire set.

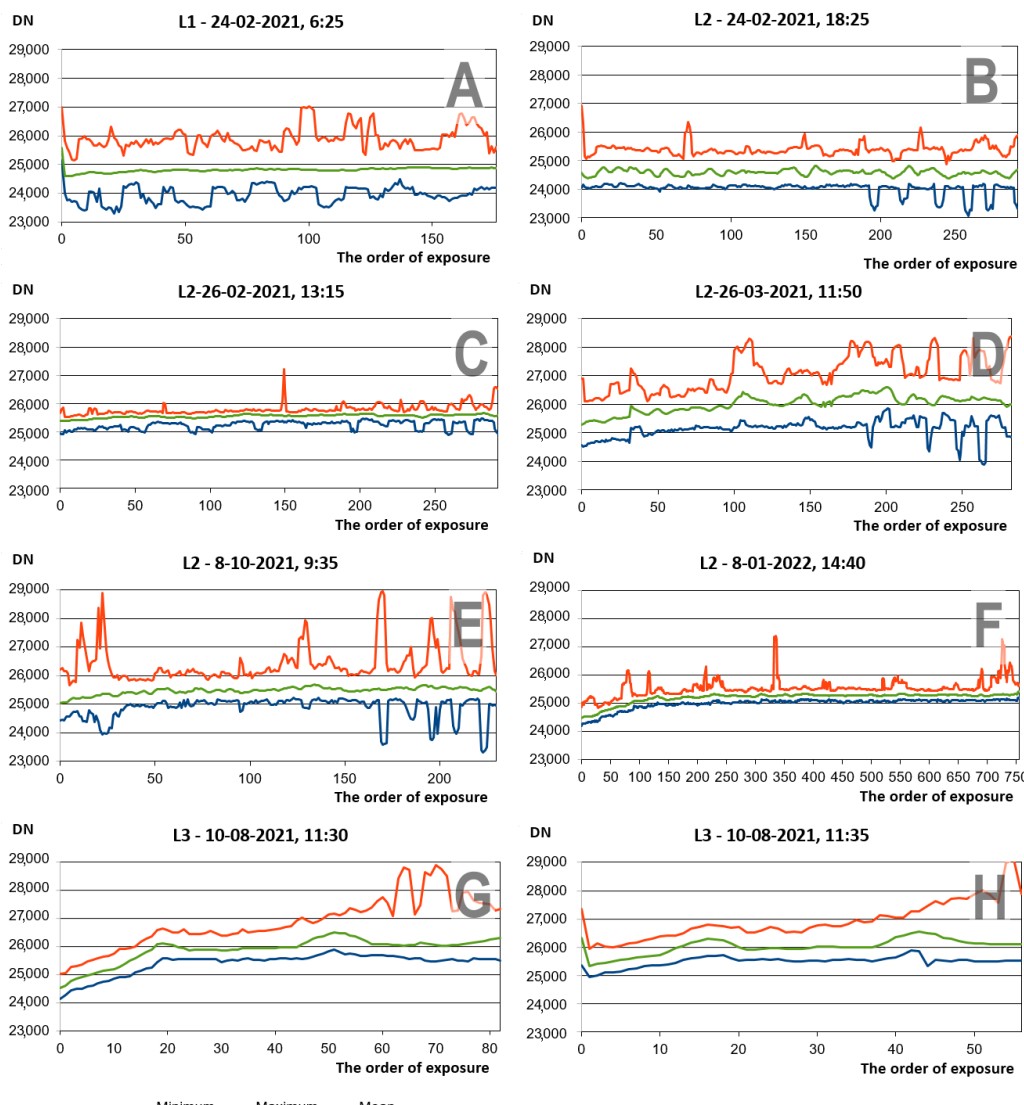

**Figure 7.** Graphs showing statistical parameters of sets including average, minimum and maximum values, of consecutive photos in individual sets according to the order of taking. The markings of the subsequent (**A–H**) charts correspond to the photo set identifiers used in Figures 5 and 6.

Figure 8 shows the reference surface temperature graphs obtained in the photos in vertical profiles up to a height of 120 m above this surface, in duplicate (profile-1: up/down/up; profile-2: up/down/up/down). Figure 9 shows the locations of the reference plot in a series of photos taken 10 to 120 m above the surface. The graphs show an increase in the reference surface temperature during the first climb. During the subsequent descent and climb steps, the measured reference surface temperature was relatively stable, within 1 degree. This slight variation in temperature can be attributed to the problem of identifying the extent of the reference surface in images of different heights and different spatial resolution.

With a single camera, it is difficult to say whether it is a defect of this unit or a property of the product, but it may be beneficial to other users of this type of camera, or to users of thermal cameras in general. The stated instability of the camera operation requires further tests (analyzes performed during subsequent raids). The solution may also be planning of the range of the photoblock so that this instability occurs outside the research area. Another idea may be to correct photos, e.g., with a Wallis filter [64].

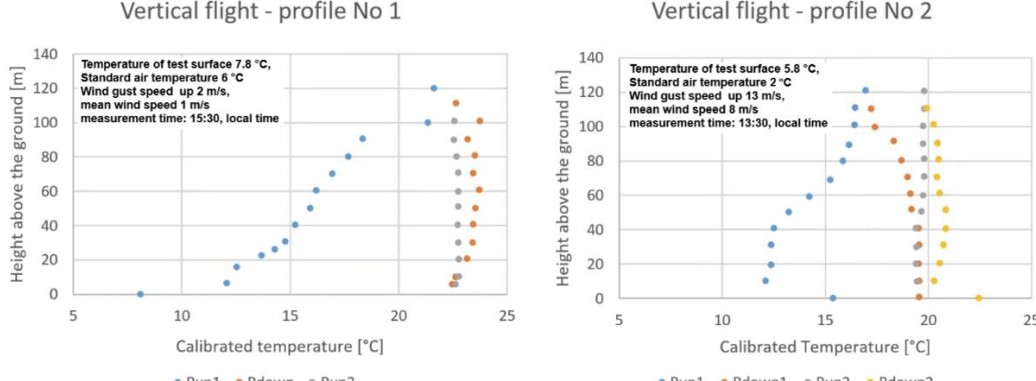

**Figure 8.** The temperature graphs of the reference surface obtained in the photos in vertical profiles up to a height of 120 m above this surface, in duplicate (profile-1: up/down/up; profile-2: up/down/up/down). The first profile was made on 4 January, and the second profile was made on 6 January 2022. For the first flight, no thermal images were taken during the second downhill series). Temperature determined by radiometric calibration equations.

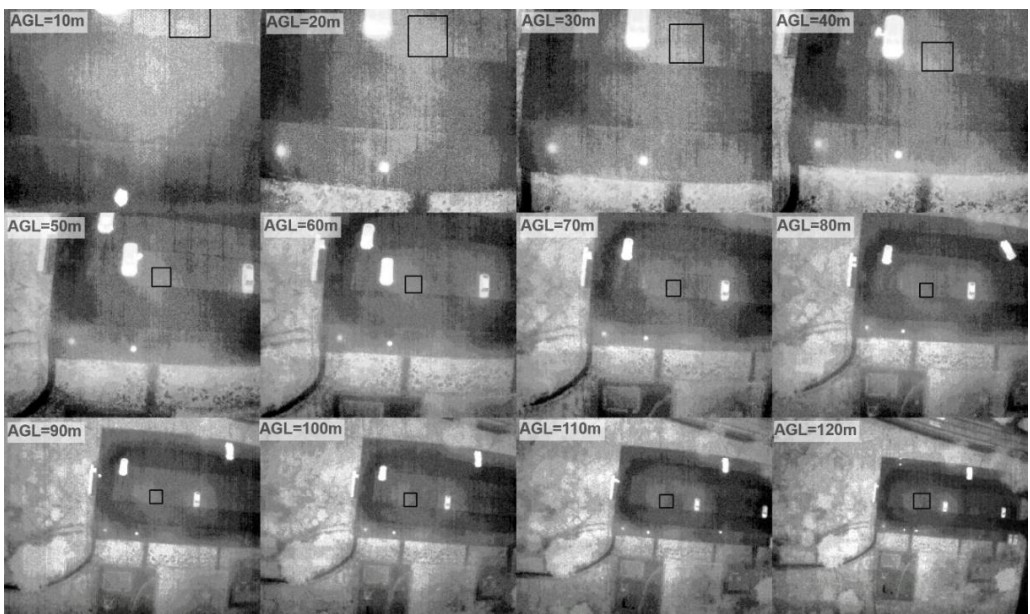

**Figure 9.** An example of a series of photos taken above the reference surface (black polygon) and the method of measuring its temperature taking into account the shrinkage of its image as in subsequent photos recorded from a higher altitude (from 10 to 120 m AGL).

## 4. Conclusions

The Yuneec E10T thermal camera, dedicated to the Yuneec H520 drone, has the possibility of practical use in quantitative research of the natural environment. Although it is sold as a product without radiometric calibration and is equipped with low-resolution matrices, the temperature measurement retains linearity. The E10T camera could be the first sensor that new users use to gain valuable experience in thermal measurement. After a simple calibration, it is possible to obtain data on the variation in temperature of natural surfaces. Performing classical photogrammetry with this camera requires appropriate planning of the height and mutual coverage of photos, in relation to the recorded content, in order to ensure the possibility of generating the necessary number of binding points in the photos during aerotriangulation. During the tests on the radiometric properties of the camera, operation instability was found in the initial period after switching on the camera. This instability can be avoided by appropriate flight planning or digitally corrected. This

stated sensor instability may be an individual case, but a more general guideline for users of remote sensing sensors may be to conduct your own tests, including analysis of image statistics or shooting parameters, especially during the warranty period. Concern for the radiometric and geometric quality of thermal data is very much from the point of view of powering geographic information systems.

**Author Contributions:** Conceptualization, A.M., S.K. and M.K. and G.J.; methodology, A.M. and S.K.; software, G.J., S.K. and A.M.; validation, M.K., A.M. and S.K.; formal analysis, A.M. and S.K.; investigation, A.M., S.K., M.K. and G.J.; resources, A.M., S.K. and G.J.; data curation, A.M., M.K. and S.K.; writing, S.K. and A.M.; writing—review and editing, A.M., S.K., M.K. and G.J.; visualization, S.K. and A.M.; supervision, S.K. and A.M.; project administration, A.M.; funding acquisition, A.M. All authors have read and agreed to the published version of the manuscript.

**Funding:** This research was funded by "UNIWERSYTET JUTRA II—zintegrowany program rozwoju Uniwersytetu im. Adama Mickiewicza w Poznaniu", no. POWR.03.05.00-00-Z303/17, co-financed by the European Social Fund under the Knowledge Education Development Operational Program (POWER) of Priority Axis III Higher Education for Economy and Development, measures 3.5 Comprehensive programs of universities.

**Data Availability Statement:** Not applicable.

**Conflicts of Interest:** The authors declare no conflict of interest. The funders had no role in the design of the study; in the collection, analyses, or interpretation of data; in the writing of the manuscript, or in the decision to publish the results.

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
