# Peer review of "Experience Gained When Using the Yuneec E10T Thermal Camera in Environmental Research"

_remotesensing, doi:10.3390/rs14112633_

Round 1

Reviewer 1 Report

The thermal imaging camera is a key tool in environmental research, and using such a camera as the payload of a UAV offers a cost-effective route towards high-performance thermal monitoring of an area of terrain. But it is essential to have a good understanding of the characteristics and the limitations of the sensor, together with the knowledge of how to obtain the best performance from it. Research in this area is plainly important and this paper contributes some useful insights and experience from the use of one type of thermal camera.

The paper is well-written with a good structure. The introduction section is interesting and useful, providing a readable introduction to the uses of thermal imaging in a wide variety of fields and clearly describing characteristics and constraints. It is particularly well-referenced. This introductory section provides background information to the E10T camera and the H520 UAV with further details appearing in section 2. This popular combination of camera and UAV is used for the experiments described in the paper. Section 2 clearly describes the methodology used - the importance of testing the calibration is described and sensible experiments are performed to assess the accuracy of the calibration. The image sets are analysed appropriately. The results are presented and analysed in section 3 - the exposition here is clear and appropriate for the task. I felt that the conclusions section was rather thin - it would be good to see some stronger statements here to help guide other users of this thermal camera.

This is a slightly unusual paper in that it describes the use of just a single camera and in effect just a single methodology. The paper's strength comes from its presentation of the importance of getting the calibration and subsequent use of the camera right. But it is desirable that the paper is not seen just as a review of one particular camera/UAV combination. Lines 420-422 are very telling and leave a feeling of concern that the use of just a single camera may cause doubt on some of the results! It's therefore important for the title to reflect this fact - maybe something like "Experience gained when using the Yuneec E10T thermal camera in environmental research" is better.

I have some minor suggestions to improve the paper - these are essentially small editorial changes:

line 2 (and line 9) - E10T -> Yuneec E10T

line 15 - dozen or even hundreds -> a large number

line 23 -  in some time the camera does not work steadily -> the camera is sometimes unstable immediately

line 25 - larger area -> area larger

line 63 - of the COVID-19 -> in the COVID-19

line 102 - lower -> lowers

line 109 - air raids -> air surveys

In Table 1 - lenght -> length
Wave range -> Wavelength
Sensitive -> Sensitivity

line 234 - made in -> made using

line 237 - can you say more about which blocks of thermal photos were selected?

line 240 - raid -> survey

line 450 was foundin -> was found in

So, in conclusion, I feel that this is an important subject for research, it has good practical relevance, the methodology is sound and interesting results are presented. But the tone of the paper should indicate to the reader that it presents experience from just a single camera and there is no evidence that all this experience has general applicability.

Author Response

Dear reviewer,

Thank you very much for your essential comments, remarks and suggestions. Referring to the following comments:

  • We agreed with the proposed change of the title of the article
  • Approved and corrected small editorial changes:
    • line 2 (and line 9) - E10T -> Yuneec E10T
    • line 15 - dozen or even hundreds -> a large number
    • line 23 -  in some time the camera does not work steadily -> the camera is sometimes unstable immediately
    • line 25 - larger area -> area larger
    • line 63 - of the COVID-19 -> in the COVID-19
    • line 102 - lower -> lowers
    • line 109 - air raids -> air surveys
    • In Table 1 - lenght -> length, Wave range -> Wavelength, Sensitive -> Sensitivity
      line 234 - made in -> made using
    • line 237 - can you say more about which blocks of thermal photos were selected?

The word “selected” was removed; about selection criteria mentioned at the purpose of the work, in further lines of text (after line 237) and results; the main aim of selection blocks was to have thermal photos taken in different areas, weather conditions, different times of the day in aspect of stable camera operation

  • line 240 - raid -> survey

  • line 450 was foundin -> was found in

We improve conclusions as whole taking into account remark that our results are based on a single instance of the device. However, modifying the content of the conclus ions, we tried to underline the important role of care for the radiometric and geometrical quality of the data.

Sincerely

Adam MÅ‚ynarczyk

Reviewer 2 Report

The paper "Various aspects of E10T thermal camera usage in environmental research" is an interesting article that proposes a method for making radiometric calibration of the E10T camera and for analyzing the statistics in order to assess the stability of the radiometric camera operation.

The paper is well-written, clear in the methodology applied by authors and also in the results. 

I just have two little comments:

1) in the figure 3 the letter A is in black colour and it is not visible in the image (Fig. 3A). I suggest to transform the letter in a light colour (white for example)

2) in the conclusions there is a sentence that needs revision.  Specifically: "During the tests on the radiometric properties of the camera, its operation instability was foundin the initial period after switching on was found".

was foundin ... was found?

Please check the sentence. 

I suppose that the sentence could be: During the tests on the radiometric properties of the camera, operation instability was found in the initial period after switching on the camera.  

Author Response

Dear reviewer,

Thank you very much for your essential comments, remarks and suggestions In Figure 3, the color of the letter A has been changed to white and all three letters have been slightly enlarged. The sentence indicated in the applications was replaced with the proposed text. We improve conclusions as whole taking into account remark that our results are based on a single instance of the device. However, modifying the content of the conclusions, we tried to underline the important role of care for the radiometric and geometrical quality of the data.

Sincerely

Adam MÅ‚ynarczyk